complexity

complex networks, air transportation network, nonlinear dynamics

**Authors for correspondence:**
Peng Ji
e-mail: pengji@fudan.edu.cn
Jürgen Kurths
e-mail: kurths@pik-potsdam.de

# The impact of COVID-19 on the worldwide air transportation network

Xiaoge Bao[1], Peng Ji[1,2], Wei Lin[2,3], Matjaž Perc[4,5,6] and Jürgen Kurths[7,8,9]

[1]Institute of Science and Technology for Brain-Inspired Intelligence, Fudan University, Shanghai 200433, People's Republic of China
[2]Research Institute of Intelligent Complex Systems and MOE Frontiers Center for Brain Science, Fudan University, Shanghai 200433, People's Republic of China
[3]School of Mathematical Sciences, LMNS, and SCMS, Fudan University, Shanghai 200433, People's Republic of China
[4]Faculty of Natural Sciences and Mathematics, University of Maribor, Koroška cesta 160, 2000 Maribor, Slovenia
[5]Department of Medical Research, China Medical University Hospital, China Medical University, Taichung, Taiwan
[6]Complexity Science Hub Vienna, Josefstädterstraße 39, 1080 Vienna, Austria
[7]Potsdam Institute for Climate Impact Research (PIK), 14473 Potsdam, Germany
[8]Department of Physics, Humboldt University, 12489 Berlin, Germany
[9]Centre for Analysis of Complex Systems, World-Class Research Center "Digital biodesign and personalized healthcare", Sechenov First Moscow State Medical University, Moscow, 119991, Russia

PJ, 0000-0002-3225-8431; WL, 0000-0002-1863-4306; MP, 0000-0002-3087-541X; JK, 0000-0002-5926-4276

Air travel has been one of the hardest hit industries of COVID-19, with many flight cancellations and airport closures as a consequence. By analysing structural characteristics of the Official Aviation Guide flight data, we show that this resulted in an increased average distance between airports, and in an increased number of long-range routes. Based on our study of network robustness, we uncover that this disruption is consistent with the impact of a mixture of targeted and random global attack on the worldwide air transportation network. By considering the individual functional evolution of airports, we identify anomalous airports with high centrality but low degree, which further enables us to reveal the underlying transitions among airport-specific representations in terms of both geographical and geopolitical factors. During the evolution of the air transportation network, we also observe how the network attempted to cope by shifting centralities between different airports around the world. Since these shifts are not aligned with optimal strategies for minimizing delays and disconnects, we conclude that they are consistent with politics trumping science from the viewpoint of epidemic containment and transport.

# 1. Introduction

On 11 March 2020, the World Health Organization declared the coronavirus disease 2019 (COVID-19) a pandemic. At that time, there were around 118 000 confirmed cases in over 110 countries and territories around the world. On 6 May, for example, that number already grew to 3 588 773 cases in over 200 countries [1,2]. This very worrying growth with the second wave is well underway all around the world [3,4]. Apart from the obvious health concerns and overburdened healthcare systems, the pandemic is wrecking havoc in industries and economies. Caixin's purchasing managers index for the services sector of China's economy fell to the lowest point in recorded history, and several stock market sectors have not been hit that hard since the ominous Black Monday in October 1987. Perhaps hardest hit [5,6].

Obviously thus, the COVID-19 pandemic is a massive global health crisis, requiring large-scale changes in behaviour and placing significant psychological burden on individuals and organizations. In this light, insights from the social and behavioural sciences can be used to help align human behaviour with the recommendations of epidemiologists and public health experts [7]. Closely monitoring and forecasting COVID-19 spreading also plays a key role to inform governments and healthcare professionals what to expect and which measures to impose, and to motivate the wider public to adhere to these measures in order to decelerate the spreading [8–11].

Quantifying and better understanding the impact of COVID-19 on different industries, on the other hand, plays a key role in further refining the containment measures and to alleviate excesses and redundancies. Here, air travel is not only one of the hardest hit industries but also a crucial factor in determining the success of containment and spreading of COVID-19. Indeed, it is responsible for the mobility of millions of people and tons of cargo every day, wielding an enormous impact on national and international economy and politics.

Seminal models that combine epidemic spreading with methods of network science and digital data have enabled us to quantify the complexity and to understand much better the key properties of spatio-temporal structures that determine epidemic spreading [12–16]. For example, the invasion threshold is strongly affected by the topological fluctuations of the underlying network, which in turn allows us to understand and foretell the effects of travel restrictions on pandemic containment [17–19]. Higher-order memory in air traffic between cities impacts epidemic spreading and reveals actual travel patterns [20]. Moreover, identifying the most efficient spreaders in a network helps to find a plausible route for an optimal design of efficient containment strategies [21]. Removing links could enhance the robustness of the network route of airlines, and analysing the evolutional properties of robustness can lead to a better understanding of the risks posed by epidemic spreading [22,23].

With this motivation, we here study the impact of COVID-19 on the worldwide air transportation network. We use data from the Official Aviation Guide, which comprise a total of 18 676 988 flights between over 3765 different airports, from 1 January to 6 May 2020. We find that cities with small degree could have anomalously large centrality and the most-connected cities are not necessarily the most central based on geographical definitions, which can be explained by the existence of airport communities [24]. We observe heterogeneous connection patterns among different airports, which is consistent with politics often trumping optimal solutions from the viewpoint of epidemic containment and transport. Evolutional connection patterns indicate the airport system's response facing this precipitate pandemic. Based on these heterogeneous connection patterns, we also determine the non-communal hierarchical structure of the worldwide air transportation network, which enables us to analyse its resilience to politically or economically inspired disruptions [25]. We use the concepts of node-specific perspectives and of effective distance to further elaborate on the impact of disruptions and to successfully explain its variability [26].

In what follows, we present the main results, firstly describing how we have constructed the worldwide air transportation network, secondly showing the effects of removed airports and flights, and finally showing how this adversely affected the evolution of global and local properties of the network. We conclude with a summary of the results and their broader implication for air travel and codependent industries.

# 2. Results

## 2.1. Network construction

For constructing worldwide air transportation networks, we study here the Official Aviation Guide (OAG) database, which is compiled by OAG Worldwide (oagdata.com). The database includes all

**Table 1.** Number of cities by major geographic region. The total number of all cities is 3750.

| region | no. locations |
| --- | --- |
| Europe | 646 |
| Asia and Middle East | 1017 |
| North America | 1107 |
| Africa | 359 |
| Oceania | 293 |
| Latin America | 328 |

operating flights and codeshare flights, and comprises flight schedule data whose airport name is uniquely identified by International Air Transport Association (IATA) code (www.iata.org).

Based on this database, we build air transportation networks consisting of cities rather than airports to simplify our analysis, and the information of one or more airports within a city is summarized in the corresponding city. The cities spatially distribute in different regions, as shown in table 1.

In order to present the evolution for air transportation networks affected by COVID-19, we focus our analysis on non-stop passenger flights for the period 1 January to 6 May 2020, and take the sliding moving windows cover across average 7 days, complying with air transport schedule cycle.

Cities are considered to be connected as long as city pairs occur once during each transport schedule cycle. City pairs are not always interconnected in reality, but no more than 5% edges are not bidirectional for each network. We consider that each connection is set as unidirectional for simplification. As we aim to show the evolution of pandemic, disappearance of part of city pairs may occur due to the effect of COVID-19 and then some cities become isolated, and we only consider the maximal connected subgraph of each network.

## 2.2. Effect of removed airports and flights

Comparing the air transportation network on the first day in 2020, the size of network, either the number of cities or city links, suffers reduction as time goes on, especially during the outbreak of COVID-19 (figure 1a,b). The minimal number of cities in figure 1a reaches 3015, and over 600 cities become isolated due to more than 10 000 links removed as shown in figure 1b compared with the city links in the first few days in 2020.

As the airline industry is barely affected in January and suffers a heavy loss in April due to aggravation of pandemic, we compare the average city/link numbers of networks in these two months. The average city number for January is 3646 and for April it is 3161. The average number of city links for January is 22 557 and for April it is 13 387, which almost halved. From the perspective of geographical locations, most of the reduced cities initially distribute in Asia and Europe as major outbreaks arise in China first at the end of January, and then the situation deteriorates. Extensive closures and cancellations spread almost evenly across all continents (figure 1c).

The aviation industry suffers greatly as seen from the intuitive results shown by increasing reduction of city number and city links. The reason is not merely a decline in need among passengers due to pandemic, but political measures such as travel restrictions also play a significant role in hit for air industry. To show the impact of pandemic on different regional aviation operations, intra-degree and inter-degree are introduced to describe domestic and foreign connecting situations, respectively. Intra-degree is the number of connections between cities inside a given region and it reveals the situation of air transportation within this region. Inter-degree is the number of edges connecting one city in this region and the other city in other regions, and it indicates the external air traffic from/to this region. After implementations of air travel bans, both intra-degree and inter-degree descend clearly for given countries as well as the European Union in figure 1d,e. The vertical dotted line labels the date for the implementation of travel restriction, pointing out the date 17 March when the European Union closed its borders and most foreign travellers are barred from entry and the date 19 March when the state department raised its global travel advisory to a Level 4, which is the agency's top warning, that US citizens either remain in place or return home [27].

Although there are obvious downward trends overall for city number, city links, intra-degree and inter-degree, there all appear uptrends after the middle of April. This indicates a recovery for airline

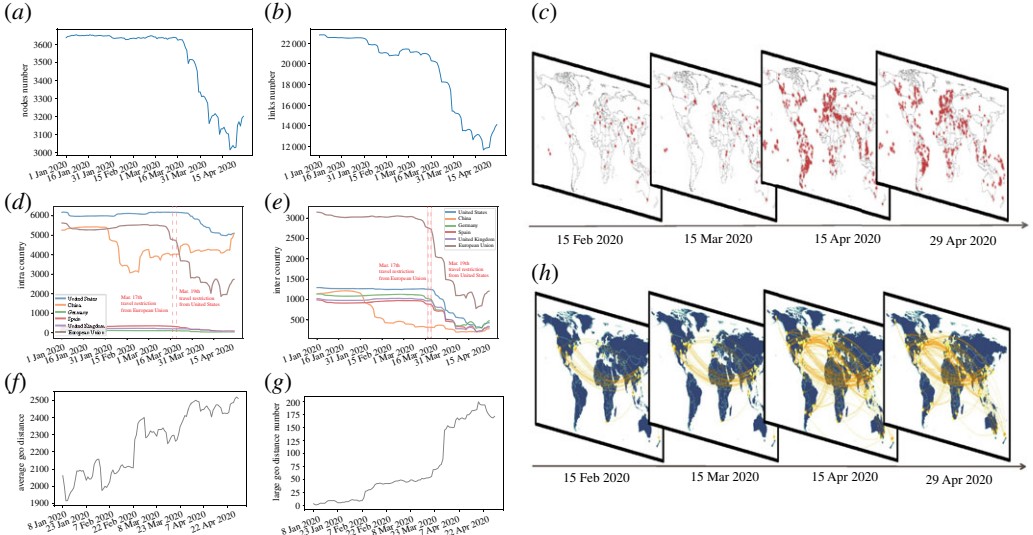

**Figure 1.** Effect of removed nodes and links. (*a*) Evolution of nodes' number in air transportation networks. (*b*) Evolution of links' number in air transportation networks. (*c*) Removed nodes comparing with the first day on worldwide maps on given days. (*d*) Evolution of intra-degree for United States, China, Germany, Spain, UK and European Union. (*e*) Evolution of inter-degree for United States, China, Germany, Spain, UK and European Union. The degree of a city is the number of adjacent cities connected by non-stop flights. Intra degree and inter degree are based on the concept of degree. Intra degree is the number of connections between cities inside the country and it reveals the situation of air transportation within that country. Inter degree is the number of edges connecting one city in the country and the other city in other countries, which indicates the external air traffic. The vertical dotted line represents date for travel restriction. (*f*) Evolution of average geographical distance for removed links comparing with the first day. For ignoring overlap of the first 7 days, the start time is 8 January 2020. (*g*) Evolution of number of removed links with large geographical distance (greater than 10 000 km) comparing with the first day. The start time is 8 January 2020 as well. (*h*) Large geographical removed links (greater than 10 000 km) comparing with the first day on worldwide maps on given days.

industry as the spread of pandemic slows down and travel bans are eased or annulled during this time in some countries, such as Russia, Canada and Denmark.

Intuitively, cancellation of flights and even shutdown of airports around the globe lead to the destruction of nodes and edges in air transportation networks. Furthermore, with regard to reduced links, the average geographical distance of these links gets longer and the number of reduced links with large geographical distance (greater than 10 000 km) increases (figure 1*f,g*). For ignoring overlap of the first 7 days, the start time is 8 January 2020 for both plots. Initially, long-range connections between China and the USA are strongly reduced, consistent with most cities being removed in China. More connections between different continents disappear afterwards, especially between Europe and North America, as well as Asia and North America (figure 1*h*). As long-range connections mainly determine the spread of disease [18], removal of links with long geographical distance effectively reduces the global scale of pandemic spreading. From more flight cancellation of long trips, there may be certain defence mechanisms to deal with the pandemic intentionally of air transportation network and internal rules behind the actual situation, which effectively control the spread of disease.

## 2.3. Evolution of global network properties

To further understand how the air transportation networks are affected by a decrease of flights and closure of airports, a conventional method is to investigate systematic changes based on network topology, and the evolution of global properties characterizing variations for networks can be employed to assess the impact of COVID-19.

Among each pair of cities, the average shortest path length reveals the efficiency of passengers transporting between cities and can further imply the potential for spread of pandemic [28]. As more and more long-range connections are removed, as shown in figure 1*g*, the average shortest path length increases accordingly, exhibited by the red dot line in figure 2*a*. Such increases were induced by multiple factors including pandemic among passengers, travel restriction, and so on. With such

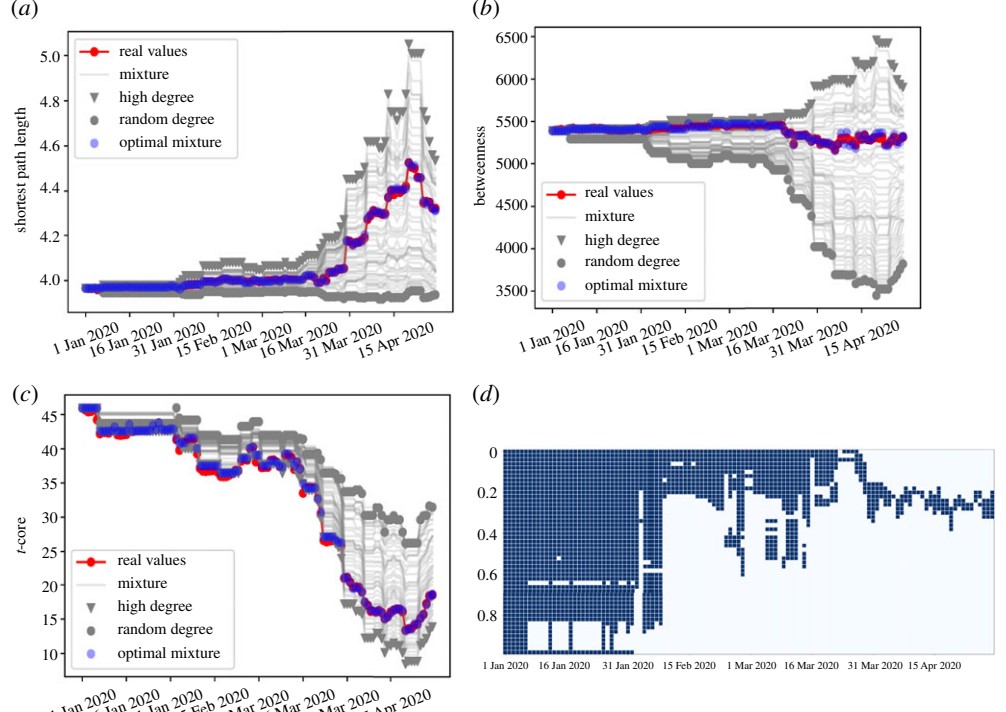

**Figure 2.** Evolution of global network properties. (*a*) Evolution for the average shortest path length under different artificial strategies and real case. (*b*) Evolution for the average betweenness under different artificial strategies and real case. (*c*) Evolution for the average *t*-core under different artificial strategies and real case. The red dots represent values in reality for each day. Nabla points are *high degree* strategy and black dots are *random* strategy. Grey lines refer to different *mixture* strategies whose parameters $\alpha$ are constants from 0 to 1 with step size set as 0.05. After varying $\alpha(t)$ according to date $t$ and going through $\alpha(t)$'s cycle of 0 to 1 whose step size is 0.02 for each $t$, the optimal series of $\alpha(t)$ for each property is obtained and each series satisfies the relative threshold $\beta$ set as 0.05. (*d*) The panel shows the mutual range of $\alpha(t)$ satisfying relative threshold set as 0.05 for the average shortest path length, betweenness and *t*-core cases on each day. The black square refers to $\alpha(t)$ on that day $t$ satisfying the relative threshold for all three properties while the white denotes the unsatisfactory situation.

increases, the average minimum number of transfers, which one needs to get from one city to others, increases. Considering the ease of travel, people less likely select long-distance trips, and this inhibits the global scale of epidemic spreading.

In additional to the shortest path length, we calculate betweenness of each city to represent its centrality [29–31]. A city with higher betweenness indicates that more flights will pass through it, and exhibits the heavy responsibility of transfer. Due to the huge disruption in the linkage of air transportation networks, the average betweenness goes up slightly at first and goes down clearly afterwards shown by red dot line in figure 2*b*. The importance of cities with vital responsibilities for transfer has been weakened.

To understand non-communal hierarchical structure of the air transportation network, we calculate the *t*-core value for each city, which was introduced in [25] (definition given in Material and methods). From the perspective of hierarchy described by the red dot line in figure 2*c*, the average *t*-core referring to the mean value of *t*-core of all cities in a network declines, indicating that the probability for cities to form triangles falls and there are less selections for passengers to arrive at their destinations.

In addition, we note that after the middle of April, there appears the opposite trend comparing the previous trend for all three real cases, which indicates the recovery of air transportation networks. This phenomenon is in accordance with figure 1*a*,*b*.

Although air industry suffers a lot and the air transportation network becomes fragile, the evolution of air transportation networks tends to the beneficial direction of pandemic control. Embodied in the evolution of global network properties from figure 2, sparser connection patterns, rendering longer path length, less betweenness and less *t*-core value, reduce the spread of disease on the world scale. To investigate the possible mechanisms for destructions behind variations of network properties for

real case, we analyse three former mentioned properties of the network by sequentially removing connections up to the given number using three different strategies. The given number of connections removed is set as $E(t)$, where $E(t)$ refers to the number of removed edges on the given day $t$. For all strategies, each subsequent step corresponds to removing up to the given number of removed edges $E(t)$.

In terms of the degree defined as the number of adjacent cities connected by non-stop flights, (i) *high degree* strategy refers to a conventional targeted removal strategy wherein each subsequent step corresponds to removal of all connections up to the given number, $E(t)$, with the highest degree; (ii) *random* strategy is that each step removes the given number of connections $E(t)$ chosen at random; (iii) *mixture* strategy is designed to balance these two ways through the parameter $\alpha$, expressed as

$$E_m(t, \alpha) = E_r(t, \alpha) + E_h(t, \alpha). \tag{2.1}$$

The number of removed edges according to *random* strategy is $E_r(t, \alpha) = E(t) * \alpha$, where $\alpha$ refers to the fraction that edges are removed under such strategy. The number of removed edges according to *high degree* strategy is $E_h(t, \alpha) = E(t) * (1 - \alpha)$, where $(1 - \alpha)$ refers to the fraction that edges are removed under *high degree* strategy. When $\alpha$ reaches 0, the mixture strategy turns into *high degree* strategy, while it transforms into *random* strategy if $\alpha$ equals to 1.

For each property, the real case fits well with *high degree* strategy shown by nabla points at the outset, but slowly closes to black dots representing *random* strategy later in figure 2*a*–*c*. Different mixture strategies are exploited through changing constant parameters $\alpha$ from 0 to 1 with step size set as 0.05 shown as grey lines in figure 2*a*–*c*.

A series of grey lines varies between *high degree* strategy and *random* strategy, and sometimes a grey line fits well with the real case in some time periods, but the overall fitting results are not good enough. For finding the ideal way to fit the real situation for a property as a whole, we vary the parameter $\alpha(t)$ according to date $t$ and design the *optimal mixture* strategy shown as blue dots in figure 2*a*–*c* which satisfies

$$\min(|F(E_m(t, \alpha(t))) - F(E(t)))|), \forall t, \tag{2.2}$$

where $F(\cdot)$ is the value of a property under a strategy. The *optimal mixture* strategies fit well with the real case for all properties exhibited in figure 2*a*–*c* but the optimal series of parameters are distinct for different properties. However, for different properties, they all satisfy the relative threshold $\beta$. For every property as mentioned, the inequality always holds

$$\frac{|F(E_m(t, \alpha(t))) - F(E(t)))|}{F(E(t))} \leq \beta. \tag{2.3}$$

Through figure 2*d*, the panel shows the range of $\alpha(t)$ satisfying relative threshold set as 0.05 for the average shortest path length, the average betweenness and the average $t$-core on each day and the mutual series of $\{\alpha(t)\}$ can be selected from the range shown by black squares. The results further figure out the intentionality for a response towards destruction of air transportation networks, and the effect of artificial factors would change along with time.

## 2.4. Evolution of local network properties

Through the evolution of global properties based on network topology, the effect of COVID-19 for air transportation networks is exhibited from the global perspective. Next, for further investigating the evolution for networks' properties more concretely, we pay attention to nodal properties, especially focusing on cities' centrality. Figure 3*a*,*b* exhibits the maximal and minimal ratio of betweenness for each city and its corresponding betweenness on the first day, indicating the huge variation of cities' betweenness. The ratio is the fraction of betweenness of a city on a day dividing that on the first day.

Betweenness of some cities with very high betweenness at the beginning are stable over time. Betweenness with big drop as time goes often points to cities with medial betweenness (1000 ∼ 10 000) at first while cities very low betweenness (less than 1) at the outset come up largely over time.

From the perspective of geographical location, cities with very high or low ratio of betweenness occupy the margin of countries most (see appendix), as they bear the vital responsibility for transfer in a certain period of time. In figure 3*b*, the city numbers, with respect to date for these cities first reaching the maximal or minimal ratio threshold, become maximum. The time for the maximal ratio and the minimal ratio of betweenness mostly concentrates in April when the worldwide network went through upheaval due to further spread of the pandemic. Hence, betweenness for the majority of cities exists as an evolutionary process and for part of cities big variations appear.

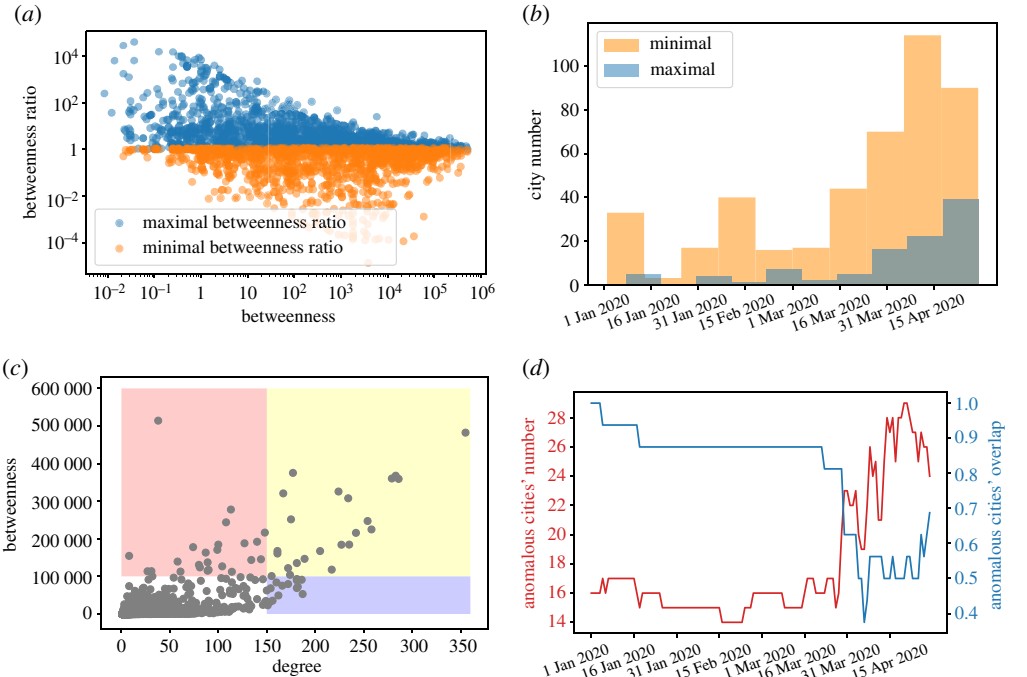

**Figure 3.** Anomalous property and compensation phenomenon. (*a*) Cities' maximal betweenness ratio and minimal betweenness ratio during the time period with respect to cities' corresponding betweenness on the first day. (*b*) City number with respect to date for these cities first reaching the maximal ratio threshold set as 100, and city number with respect to date for these cities first reaching the minimal ratio threshold set as 0.01. (*c*) Cities for degree and betweenness on the first day in 2020. The graph is divided into four parts. (*d*) Anomalous city number (degree less than 100, betweenness more than 100 000) versus time and the overlap number of anomalous cities between that day and the first day.

As part of cities with high betweenness have sharply dropping betweenness, we are curious about these cities' evolution of degree as well, because these two nodal properties can show the importance of cities in air transportation networks from different perspectives. Firstly, the first day's nodal properties show that not all central cities are highly connected in figure 3*c* and these cities with low degree but high betweenness are vital in evolution whose importance is often overlooked. In reality, cities with low degree may be destroyed easily as these cities serve the need of minority, and airline companies tend to cut down the relevant airlines considering economic benefits and other factors during pandemic. Although these cities' degree is not very high, they undertake an important responsibility of transfer and play a key role in phenomena such as diffusion and congestion due to high betweenness. Once airports of these cities close down, the average betweenness of the whole network might bear the brunt. Nodes with low degree but high betweenness are called anomalous nodes in [24].

The unexpected finding of anomalous nodes is a very important one, because central nodes play a crucial part in the network, but the property of less connected makes these nodes vulnerable in face of pandemic. For example, Tamuin, a municipio in the Mexican state of San Luis Potosi, had a very high betweenness in March but became isolated in the whole April as its connections were removed. The evolution for the number and overlap of anomalous nodes in figure 3*d* indicates that anomalous nodes may convert to other types, whereas part of nodes may become anomalous nodes over time. Not all cities being anomalous at the very start can maintain anomalous all the time. Betweenness of part of cities with low degree initially increases and then these cities become anomalous as time goes on. During the later period, cities with both high degree and betweenness have declining degree to be anomalous over time. The anomalous property shifts along with time and we name this phenomenon *compensation*. For further investigating compensation through some targeted cities, table 2 exhibits cities with the top 25 betweenness on the first day in 2020 and puts the corresponding degree and existence which indicates the compensation phenomenon. Existence means the days of this city still being among the top 25 central cities during the time period dividing the length of cycle which is 120 days.

In table 2, Anchorage has very high betweenness but its degree is low comparatively, as well as Brisbane and Sydney. Although these cities have fewer direct connections with other cities, they bear

**Table 2.** Top 25 cities with the highest betweenness and their degree on 1 January 2020. Existence is the number of days the corresponding city has been in the top 25 most central cities, divided by the length of cycle which is 120 days.

| rank | cities | betweenness | degree | existence |
|------|--------|-------------|--------|-----------|
| 1 | USA–Anchorage | 514093.7 | 38 | 1.00 |
| 2 | GB–London | 482179.9 | 355 | 1.00 |
| 3 | USA–Los Angeles | 375143.5 | 177 | 1.00 |
| 4 | TR–Istanbul | 367631.9 | 283 | 0.93 |
| 5 | RU–Moscow | 360263.6 | 279 | 1.00 |
| 6 | FR–Paris | 359173.3 | 286 | 1.00 |
| 7 | AE–Dubai | 325621.7 | 224 | 0.74 |
| 8 | JP–Tokyo | 320679.2 | 167 | 1.00 |
| 9 | USA–Chicago | 307664.3 | 234 | 0.99 |
| 10 | USA–Seattle | 277730.6 | 113 | 1.00 |
| 11 | CA–Toronto | 251607.1 | 175 | 1.00 |
| 12 | CN–Beijing | 247193.5 | 254 | 0.93 |
| 13 | BR–Sao Paulo | 243714.5 | 108 | 0.97 |
| 14 | DE–Frankfurt am Main | 224842.7 | 258 | 0.73 |
| 15 | SG–Singapore | 216441.1 | 148 | 0.98 |
| 16 | NL–Amsterdam | 215739.5 | 242 | 1.00 |
| 17 | DK–Copenhagen | 192082.7 | 140 | 0.07 |
| 18 | CA–Montreal | 187962.5 | 127 | 0.70 |
| 19 | CA–Vancouver | 184462.7 | 100 | 0.75 |
| 20 | USA–Dallas-Fort Worth | 184446.5 | 235 | 0.94 |
| 21 | CN–Shanghai | 184155.9 | 227 | 0.67 |
| 22 | AU–Brisbane | 177729.6 | 74 | 0.37 |
| 23 | AU–Sydney | 170599.9 | 99 | 0.83 |
| 24 | KR–Seoul | 167524.9 | 161 | 0.87 |
| 25 | USA–NY | 167350.2 | 205 | 1.00 |

the duty of transfer and are significant for connectivity of network and other properties. Seen from table 2, the betweenness of these cities is not always very high over time through existence, and the anomalous property of part of cities shifts along with time, which reflects compensation. To further explain the cause of this phenomenon, we mainly focus on cities' centrality and integrate top 25 central cities for each day, and consequently there are 40 cities appearing during the whole period. Except for cities being top central all the time whose existence in table 2 equal to zero, figure 4*a* shows betweenness variation along with time. Part of vital cities' betweenness changes dramatically during the period of outbreak and they may reflect effect of economical and political factors due to the pandemic. Among these cities, there are four cities identified whose betweenness decreases to 50 per cent on the last day comparing with the first day and increases to 50 per cent respectively (see appendix).

Cities with decreasing betweenness also have decreasing degree and the drop of degree is more than 50 per cent comparing with that on the first day. In addition, we find that these four cities (including Dubai, Montreal, Frankfurt am Main and Jeddah) are famous for tourism whose betweenness and degree are high at the start. Because of pandemic, tourist attraction suffers a blow and its importance in the network reduces. In addition, we find degree of cities with increasing betweenness decreases and then increases, and the variation is slight comparing with cities with dropping betweenness. These four cities, consisting of Abu Dhabi, Addis Ababa, Mexico City and Panama City, are all the capital of their own country. Comparing the former four cities, the key role in air transportation evolution is from tourism economy to politics due to COVID-19. The phenomenon not only reflects the evolution of anomalous properties for nodes over time, but also implies the migration of economic and political factors under the COVID-19. From figure 4*b*, the migration is shown from the perspective of geographical locations.

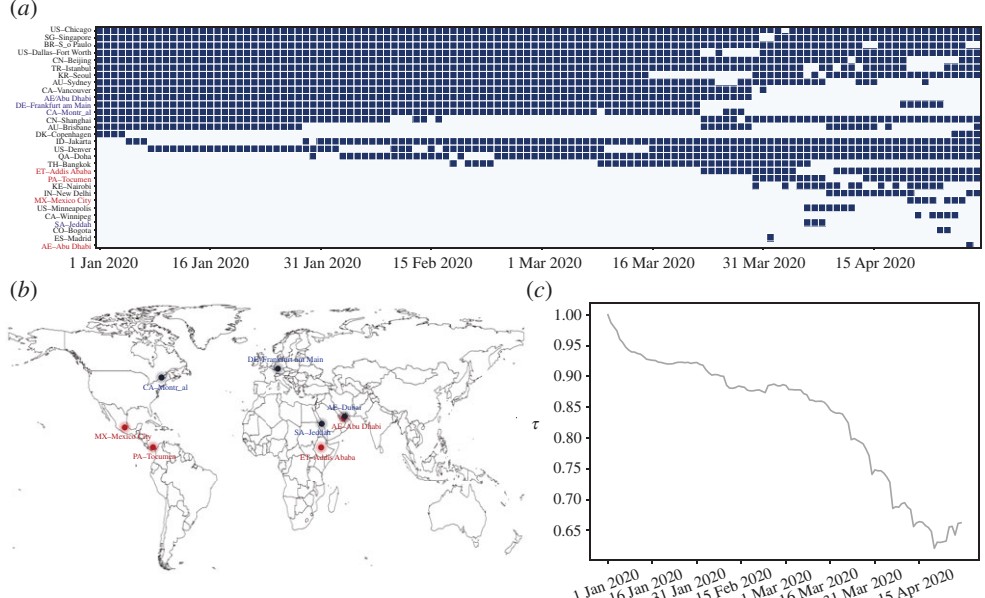

**Figure 4.** Centrality variation for cities with high betweenness. (*a*) Cities' evolution according to rankings of betweenness. Black square represents the city is one of the 25 most central cities on that day, while the white denotes the city is not in the top central cities on that day. Identify 25 most central cities on each day, and there are a total of 40 cities appearing during the period we study. Except for 10 cities whose existence equal to one in table 2, which means they are always top central cities during the period, we show the betweenness evolution for the remainder. Existence in table 2 is also the number of black squares for a city occupying its total squares. (*b*) Cities whose betweenness decreases to 50 per cent on the last day compared with the first day and increases to 50 per cent on the worldwide map. Red dots are cities with increasing betweenness which are political centres, while dark-blue dots are tourist cities with decreasing betweenness. (*c*) Evolution of correlation $\tau$. $\tau$ is the correlation between the vector of cities' rankings in terms of betweenness on the first day and that on a given day. These cities always exist in networks during the whole period and have non-zero betweenness on the first day.

For more general purpose to show the evolution of betweenness and the cause of compensation, we consider cities, which exist during the whole period and have non-zero betweenness on the first day. These cities have different betweenness on each day and their rankings according to betweenness also vary over time. The decreasing correlation $\tau$ between rankings on the first day and on a given day reflects more and more drastic transformation for nodal centrality. The upswing in the later period indicates the recovery of betweenness. We additionally provide effect of removed airports and flights, evolution of global network properties, and other corresponding information in the online electronic supplementary material.

## 3. Discussion and conclusion

We have studied the impact of COVID-19 on the worldwide air transportation network. Air travel has been one of the hardest hit industries of COVID-19, and it also critically determines the success of containment and spreading of epidemics as it is responsible for the mobility of people and cargo around the world. Our research shows that during the height of the first wave of COVID-19 more than 600 out of over 3700 airports were shut down, the number of flights decreased by 50%, the average distance between airports increased by a quarter, and the number of removed flights longer than 10 000 km increased more than 20-fold. In simulating such outages by means of random failures and targeted attack [32], we show that the COVID-19 disruption of the worldwide air transportation network is consistent with the impact of a mixture of targeted and random attacks. We also reveal how the network yields an enormous impact on national and international economy and politics, which is manifested by the shifting centralities between different airports around the world. Such shifts do not seem to be guided by optimization and the mitigation of adverse effects, but rather by seemingly arbitrary shutdowns with a political background.

As for the targeted impact of epidemic containment, our research shows that the number of flights and operational airports has decreased steadily from January onwards, as a consequence having significantly larger average geographical distances. The world thus effectively got much bigger, and since long-range connections significantly shrunk the travel time, the cancellation of long-range

connections curbs the population flow and helps to reduce the global scale of epidemic spreading. We have observed that properties characterizing the network varied most significantly from mid-March to mid-April, when the connections between many major cities around the world became weak and when passengers thus faced many more transfers to reach destinations.

We have shown that some airports with a small degree have anomalous properties, and thus despite their seemingly marginal character they play a key role in the evolution of the air transportation network. Anomalous nodes—nodes with low degree but high betweenness—have a significant influence on the evolution of the average betweenness. These nodes have a marked compensation, reflecting the fact that anomalous nodes may rapidly convert and loose their status while other nodes take up their roles. Affected by the COVID-19 pandemic, the compensation phenomenon is more obvious due to the significant variation of the betweenness and the degree for some cities. In particular, we find that cities with initial high betweenness and descending betweenness afterwards are typically popular tourist destinations, while cities with increasing betweenness are political centres. Moreover, we have observed recoveries in the later periods of our study due to the overall recovery of the air transportation network.

We hope our research will be useful to devise more efficient policies for mitigating adverse effects of flight and airport shutdowns in future pandemics, or as most recent data indicate, during the second wave of the COVID-19 pandemic. Much like lethality and centrality in protein networks have been used to learn that the most highly connected proteins in the cell are the most important for its survival [33], so could research of the air transportation network reveal which flights and airports are crucial for sustained services and which could be shutdown without causing major disruptions. Of course, these considerations come second only to primary data about infection hubs and active cases, but when possible, they could and should be used to trump political agenda.

## 4. Material and methods

### 4.1. Air flight data

Air flight data are commercially available from the Official Aviation Guide (OAG, www.oag.com) and the International Air Transport Association (IATA, www.iata.org) database. OAG is the world's leading provider of digital flight information and contains information about flight schedules and frequencies for airlines throughout the world, including air carrier with IATA codes, origin airport, destination airport, seats, frequency and so on. We use the air flight data from 1 January to 6 May 2020, obtain 18 676 988 flights in total and over 3765 different airports during this period. We obtain the geographical coordinates of the world airports and airline company name from the IATA database.

### 4.2. Definition of the *t*-core decomposition

Similar to the *k*-core method, the *t*-core decomposition first removes all nodes that are not part of any triangles [25]. From small to large, nodes less than or equal to the number of triangles they form are removed. At each iteration, the nodes that are removed correspond to the current number of triangles they constitute, namely the *t*-core.

Data accessibility. Proprietary airline data are commercially available from the OAG (www.oag.com) and IATA (www.iata.org) databases. Air flight data are commercially available from the Official Aviation Guide.
Authors' contributions. P.J., W.L., M.P. and J.K. designed the research. P.J. and X.B. collected the data and implemented the model. All authors contributed analysis ideas and wrote the paper.
Competing interests. We declare we have no competing interests.
Funding. This work was supported by Shanghai Municipal Science and Technology Major Project (grant no. 2021SHZDZX0103), National Natural Science Foundation of China (grant no. 62076071), and the Slovenian Research Agency (grant nos. P1-0403 and J1-2457). JK was financed by the Ministry of Science and Higher Education of the Russian Federation within the framework of state support for the creation and development of World-Class Research Centers "Digital biodesign and personalized healthcare" No. 075-15-2020-926.

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
