## [Peer Review File · Royal Society Open Science]

Review History

RSOS-210682.R0 (Original submission)

Review form: Reviewer 1

Is the manuscript scientifically sound in its present form?

Yes

Are the interpretations and conclusions justified by the results?

Yes

Is the language acceptable?

Yes

Do you have any ethical concerns with this paper?

No

Have you any concerns about statistical analyses in this paper?

No

Recommendation?

Accept with minor revision (please list in comments)

Comments to the Author(s)

In their work entitled “The impact of COVID-19 on the worldwide air transportation network”, the authors analyse data from the Official Aviation Guide database and build a worldwide air transportation network whose evolution, during the outbreak of COVID-19 (i.e., in the first months of 2020), is investigated. First, the authors perform a descriptive analysis where they depict how observables such as number of links, degrees, distance among nodes, vary in time. Then, the authors address more involved analyses such as an estimate of the network resilience to politically or economically inspired disruptions.

In general, I find that the work is well motivated and results are of interest for a broad audience. I have only a few minor points that I would like the authors to consider before I can recommend publication.

1. The first paragraph of the introductory section should be updated, in particular, sentences like “there are still no signs that this very worrying growth is decelerating with the 2nd wave well underway in the U.S. and in Europe” look inconsistent with the current scenario.
2. As I wrote above, part of the analysis aims to unveil intrinsic properties of the network and their role compared to political or economical actions, or compared to the spreading of the epidemics. In my opinion, this is the most interesting part of the paper, but it requires some revisions to make it more easily accessible to the reader. For instance, the paragraph after table 1, or the second paragraph in the section “Evolution of global network properties” need to be rewritten. Sentences like “the result verifies again that the evolution of air transportation networks in real case tends to contain infection diseases” should be more clearly motivated; also, statement like “... since long-range connections facilitate epidemics, it may well be that the measures nonetheless reduced the global scale of epidemic spreading” should be accurately explained.
3. After eq. 1, please check “upper bound 0” and “lower bound 1”.
4. I appreciate that technical details are collected in the Supplementary Material as this makes the main text much fluent. However, I suggest that the authors introduce minimal definitions for the quantities depicted in Figs. 2 and 3, because in some cases it is difficult for the reader to grasp the actual content of the figures and therefore the message conveyed.

Review form: Reviewer 2

Is the manuscript scientifically sound in its present form?

Yes

Are the interpretations and conclusions justified by the results?

Yes

Is the language acceptable?

Yes

Do you have any ethical concerns with this paper?

No

Have you any concerns about statistical analyses in this paper?

No

Recommendation?

Major revision is needed (please make suggestions in comments)

Comments to the Author(s)

This paper studies the impact of the COVID-19 pandemic on the global air transport network . The authors analyze the structural characteristics of the flight data and show that this resulted in increased average distance between airports and in an increased number of long-range routes . Based on their analysis , the authors conclude that this disruption is consistent with a mixture of targeted and random global attack on the worldwide air transportation network . By considering the individual functional evolution of airports , they identify anomalous airports with high-centrality but low-degree , which further enables to reveal the underlying transitions among airport-specific representations in terms of both geographical and geopolitical factors . The paper is well-written and easy to follow . However , I have the following concerns . 1 .The authors claim that this paper is the first one to study the network robustness , but it is not clear to me how the network is robust to the attacks . For example , in Figure 2 (a) , the average shortest path length increases exhibited by the red dot line in Fig. 1 (g) . The reason is not merely a decline in need among passengers due to pandemic , but political measures such as travel restric- tions also play a significant role in hit for air industry . The results further figure out the intentionality for a response towards destruction of air transportation networks , and the effect of artificial factors would change along with time . 2 .In Figure 3 (b) , it seems that the time for the maximal ratio and the minimal ratio of betweenness mostly concentrates in April when the worldwide network went through upheaval due to further spread of the pandemic . The finding of anomalous nodes is a very important one because central nodes play crucial role but the property of these nodes vulnerable in pandemic is not known . 3 .In Table 1 , the correlation between correlation between the first day and more drastic transformation on first day reflects more dramatic transformation on a given day . 4 .In the last paragraph of Section 3.1 , it is mentioned that the average city/link numbers of networks in these two months . It would be better if the authors can provide more details about the number of cities .

Decision letter (RSOS-210682.R0)

Dear Dr Ji,

The Editors assigned to your paper RSOS-210682 "The impact of COVID-19 on the worldwide air transportation network" have now received comments from reviewers and would like you to revise the paper in accordance with the reviewer comments and any comments from the Editors. Please note this decision does not guarantee eventual acceptance.

Please submit your revised manuscript and required files (see below) no later than 21 days from today's (ie 16-Aug-2021) date. Note: the ScholarOne system will 'lock' if submission of the revision is attempted 21 or more days after the deadline. If you do not think you will be able to meet this deadline please contact the editorial office immediately.

on behalf of Dr Feng Fu (Associate Editor) and Mark Chaplain (Subject Editor)
openscience@royalsociety.org

Reviewer comments to Author:

Reviewer: 1

Comments to the Author(s)

In their work entitled "The impact of COVID-19 on the worldwide air transportation network", the authors analyse data from the Official Aviation Guide database and build a worldwide air transportation network whose evolution, during the outbreak of COVID-19 (i.e., in the first months of 2020), is investigated. First, the authors perform a descriptive analysis where they depict how observables such as number of links, degrees, distance among nodes, vary in time. Then, the authors address more involved analyses such as an estimate of the network resilience to politically or economically inspired disruptions.

In general, I find that the work is well motivated and results are of interest for a broad audience. I have only a few minor points that I would like the authors to consider before I can recommend publication.

1. The first paragraph of the introductory section should be updated, in particular, sentences like "there are still no signs that this very worrying growth is decelerating with the 2nd wave well underway in the U.S. and in Europe" look inconsistent with the current scenario.
2. As I wrote above, part of the analysis aims to unveil intrinsic properties of the network and their role compared to political or economical actions, or compared to the spreading of the epidemics. In my opinion, this is the most interesting part of the paper, but it requires some revisions to make it more easily accessible to the reader. For instance, the paragraph after table 1, or the second paragraph in the section "Evolution of global network properties" need to be rewritten. Sentences like "the result verifies again that the evolution of air transportation

networks in real case tends to contain infection diseases" should be more clearly motivated; also, statement like "... since long-range connections facilitate epidemics, it may well be that the measures nonetheless reduced the global scale of epidemic spreading" should be accurately explained.

3. After eq. 1, please check "upper bound 0" and "lower bound 1".

4. I appreciate that technical details are collected in the Supplementary Material as this makes the main text much fluent. However, I suggest that the authors introduce minimal definitions for the quantities depicted in Figs. 2 and 3, because in some cases it is difficult for the reader to grasp the actual content of the figures and therefore the message conveyed.

Reviewer: 2

Comments to the Author(s)

This paper studies the impact of the COVID-19 pandemic on the global air transport network . The authors analyze the structural characteristics of the flight data and show that this resulted in increased average distance between airports and in an increased number of long-range routes . Based on their analysis , the authors conclude that this disruption is consistent with a mixture of targeted and random global attack on the worldwide air transportation network . By considering the individual functional evolution of airports , they identify anomalous airports with high-centrality but low-degree , which further enables to reveal the underlying transitions among airport-specific representations in terms of both geographical and geopolitical factors . The paper is well-written and easy to follow . However , I have the following concerns . 1 .The authors claim that this paper is the first one to study the network robustness , but it is not clear to me how the network is robust to the attacks . For example , in Figure 2 (a) , the average shortest path length increases exhibited by the red dot line in Fig. 1 (g) . The reason is not merely a decline in need among passengers due to pandemic , but political measures such as travel restric- tions also play a significant role in hit for air industry . The results further figure out the intentionality for a response towards destruction of air transportation networks , and the effect of artificial factors would change along with time . 2 .In Figure 3 (b) , it seems that the time for the maximal ratio and the minimal ratio of betweenness mostly concentrates in April when the worldwide network went through upheaval due to further spread of the pandemic . The finding of anomalous nodes is a very important one because central nodes play crucial role but the property of these nodes vulnerable in pandemic is not known . 3 .In Table 1 , the correlation between correlation between the first day and more drastic transformation on first day reflects more dramatic transformation on a given day . 4 .In the last paragraph of Section 3.1 , it is mentioned that the average city/link numbers of networks in these two months . It would be better if the authors can provide more details about the number of cities .

===PREPARING YOUR MANUSCRIPT===

===PREPARING YOUR REVISION IN SCHOLARONE===

<https://royalsociety.org/journals/authors/author-guidelines/#supplementary-material> to include a suitable title and informative caption. An example of appropriate titling and captioning may be found at https://figshare.com/articles/Table_S2_from_Is_there_a_trade-off_between_peak_performance_and_performance_breadth_across_temperatures_for_aerobic_scooping_in_teleost_fishes_/3843624.

Author's Response to Decision Letter for (RSOS-210682.R0)

See Appendix A.

RSOS-210682.R1 (Revision)

Review form: Reviewer 1

Is the manuscript scientifically sound in its present form?

Yes

Are the interpretations and conclusions justified by the results?

Yes

Is the language acceptable?

Yes

Do you have any ethical concerns with this paper?

No

Have you any concerns about statistical analyses in this paper?

No

Recommendation?

Accept as is

Comments to the Author(s)

The authors have satisfactorily addressed all the points raised in my previous report. I am happy to recommend this paper for publication in the Royal Society Open Science.

Decision letter (RSOS-210682.R1)

Dear Dr Ji,

It is a pleasure to accept your manuscript entitled "The impact of COVID-19 on the worldwide air transportation network" in its current form for publication in Royal Society Open Science. The comments of the reviewer(s) who reviewed your manuscript are included at the foot of this letter.

COVID-19 rapid publication process:

We are taking steps to expedite the publication of research relevant to the pandemic. If you wish, you can opt to have your paper published as soon as it is ready, rather than waiting for it to be published the scheduled Wednesday.

This means your paper will not be included in the weekly media round-up which the Society sends to journalists ahead of publication. However, it will still appear in the COVID-19 Publishing Collection which journalists will be directed to each week (<https://royalsocietypublishing.org/topic/special-collections/novel-coronavirus-outbreak>).

If you wish to have your paper considered for immediate publication, or to discuss further, please notify openscience_proofs@royalsociety.org and press@royalsociety.org when you respond to this email.

on behalf of Dr Feng Fu (Associate Editor) and Mark Chaplain (Subject Editor)
openscience@royalsociety.org

Reviewer comments to Author:
Reviewer: 1

Comments to the Author(s)
The authors have satisfactorily addressed all the points raised in my previous report. I am happy to recommend this paper for publication in the Royal Society Open Science.

Appendix A

Dear Prof. Fu and Prof. Chaplain,

Thanks a lot for providing these useful comments. Following the referees' suggestions, we have modified the manuscript accordingly and have replied to each of their suggestions.

Best wishes,

Xiaoge Bao, Peng Ji, Wei Lin, Matjaž Perc, and Jürgen Kurths

1 Reviewer: 1

In their work entitled “The impact of COVID-19 on the worldwide air transportation network”, the authors analyse data from the Official Aviation Guide database and build a worldwide air transportation network whose evolution, during the outbreak of COVID-19 (i.e., in the first months of 2020), is investigated. First, the authors perform a descriptive analysis where they depict how observables such as number of links, degrees, distance among nodes, vary in time. Then, the authors address more involved analyses such as an estimate of the network resilience to politically or economically inspired disruptions.

In general, I find that the work is well motivated and results are of interest for a broad audience. I have only a few minor points that I would like the authors to consider before I can recommend publication.

We appreciate the overall positive assessment of our manuscript. We answer each of your comments as follows:

Comment 1:

The first paragraph of the introductory section should be updated, in particular, sentences like “there are still no signs that this very worrying growth is decelerating with the 2nd wave well underway in the U.S. and in Europe” look inconsistent with the current scenario.

Response 1:

Thanks for pointing out our linguistic inaccuracies. The sentence you pointed out indeed caused ambiguity. We have updated the corresponding sentences according to the current scenario. We have updated the first paragraph of the introductory section in the revision of the manuscript.

Comment 2:

As I wrote above, part of the analysis aims to unveil intrinsic properties of the network and their role compared to political or economical actions, or compared to the spreading of the epidemics. In my opinion, this is the most interesting part of the paper, but it requires some revisions to make it more easily accessible to the reader. For instance, the paragraph after table 1, or the second paragraph in the section “Evolution of global network properties” need to be rewritten. Sentences like “the result verifies again that the evolution of air transportation networks in real case tends

to contain infection diseases” should be more clearly motivated; also, statement like “... since long-range connections facilitate epidemics, it may well be that the measures nonetheless reduced the global scale of epidemic spreading” should be accurately explained.

Response 2:

Thanks for your suggestions. We fully agree with your comments. Following your suggestions, we rewrite the second paragraph in the section “Evolution of global network properties” in this revision. Additionally, we also amend other paragraphs of this section.

Comment 3:

After eq. 1, please check “upper bound 0” and “lower bound 1”.

Response 3:

Thanks for pointing out our error in this expression. Here, the upper bound of Eq. (1) is 1, and the lower bound is 0. To avoid the error in the expression and make it easier to follow, we correct this expression in the revision of the manuscript.

Comment 4:

I appreciate that technical details are collected in the Supplementary Material as this makes the main text much fluent. However, I suggest that the authors introduce minimal definitions for the quantities depicted in Figs. 2 and 3, because in some cases it is difficult for the reader to grasp the actual content of the figures and therefore the message conveyed.

Response 4:

Thanks for your suggestions. We indeed introduced many quantities, e.g., Random strategy, Mixture strategy, and so on, depicted in Figs. 2 and 3, and interpreted these quantities both in the captions and in the main text. We have removed duplicated definition of these quantities in the caption of Figs. 2 and Fig. 3. To make readers grasp the key points easier, we have also updated the caption of Figs. S2 and S3 in the Supplementary Material.

2 Reviewer: 2

This paper studies the impact of the COVID-19 pandemic on the global air transport network. The authors analyze the structural characteristics of the flight data and show that this resulted in increased average distance between airports and in an increased number of long-range routes. Based on their analysis, the authors conclude that this disruption is consistent with a mixture of targeted and random global attack on the worldwide air transportation network. By considering the individual functional evolution of airports, they identify anomalous airports with high-centrality but low-degree, which further enables to reveal the underlying transitions among airport-specific representations in terms of both geographical and geopolitical factors. The paper is well-written and easy to follow. However, I have the following concerns.

We appreciate the overall positive assessment of our manuscript. We answer each of your comments as follows:

Comment 1:

The authors claim that this paper is the first one to study the network robustness, but it is not clear to me how the network is robust to the attacks. For example, in Figure 2 (a) , the average shortest path length increases exhibited by the red dot line in Fig. 1 (g) . The reason is not merely a decline in need among passengers due to pandemic, but political measures such as travel restrictions also play a significant role in hit for air industry. The results further figure out the intentionality for a response towards destruction of air transportation networks, and the effect of artificial factors would change along with time.

Response 1:

Sorry for these misleading points. We are not intended to have this claim. This manuscript is not the first one to study network robustness. We included some previous work on network robustness in the section of the introduction. The evolutionary curve for Fig. 1(g) refers to the number of removed links with large geographical distance ($> 10,000$ km) based on the network whose link weight is the geographical distance, but the red dot line in Fig. 2(a) refers to the shortest path length corresponding to the network with link weight which is the average of the nonstop passenger flights. These two evolutionary curves have the same tendencies and convey a part of the same information, which both indicate the intentionality of the network. We agree with your points that the average shortest path length increases due to multiple reasons, including pandemic among passengers, travel restriction by country, and so on. The evolution of various properties is only one straightforward way to represent the airport networks' response.

Comment 2:

In Figure 3 (b) , it seems that the time for the maximal ratio and the minimal ratio of betweenness mostly concentrates in April when the worldwide network went through upheaval due to further spread of the pandemic. The finding of anomalous nodes is a very important one because central nodes play crucial role but the property of these nodes vulnerable in pandemic is not known .

Response 2:

Yes, indeed, as shown in Fig. 3(b), the city numbers, with respect to date for these cities first reaching the maximal or minimal ratio threshold, become maximum in April due to the further spread of the pandemic. We have updated the corresponding sentences in the section of "Evolution of local network properties" and in the section of "Discussion and Conclusions" in the revision.

The vulnerable property of these anomalous nodes is mainly reflected by their small degrees, and thus when more links disappear, they tend to be isolated compared to nodes with large degree. The finding of anomalous nodes is very important because central nodes play crucial role in the passenger transfer. We have also provided further properties on these cities in the section of "Evolution of local network

properties”. For example, Tamuín, a municipio in the Mexican state of San Luis Potosí, had very high betweenness in March but became isolated in the whole April as its connections were removed.

Comment 3:

In Table 1, the correlation between correlation between the first day and more drastic transformation on first day reflects more dramatic transformation on a given day.

Response 3:

We agree with your opinion, and the drastic variation of evolutionary correlation in Fig. 4 reflects dramatic transformation on the given day.

Comment 4:

In the last paragraph of Section 3.1 , it is mentioned that the average city/link numbers of networks in these two months . It would be better if the authors can provide more details about the number of cities .

Response 4:

Thanks for your suggestions. The minimal number of cities in Fig. 1 (a) reaches 3,015, and over 600 cities become isolated due to more than 10,000 links removed as shown in Fig. 1 (b) compared to the city links in the first few days in 2020. In the section “Effect of removed airports and flights”, we have provided more details about the number of cities and links to illustrate the effect of pandemic in the second paragraph.